# An Optical Technique to Produce Embedded Quantum Structures in Semiconductors

**DOI:** 10.3390/nano13101622

**Published:** 2023-05-12

**Authors:** Cyril Hnatovsky, Stephen Mihailov, Michael Hilke, Loren Pfeiffer, Ken West, Sergei Studenikin

**Affiliations:** 1Emerging Technologies Division, National Research Council of Canada, Ottawa, ON K1A 0R6, Canada; 2Department of Physics, McGill University, Montreal, QC H3A 2T8, Canada; 3Department of Electrical Engineering, Princeton University, Princeton, NJ 08544, USA

**Keywords:** quantum structures, structured light, lateral superlattice, embedded nano-structures, Weiss oscillations, commensurability oscillations, photo-doping, persistent photoconductivity, AlGaAs

## Abstract

The performance of a semiconductor quantum-electronic device ultimately depends on the quality of the semiconductor materials it is made of and on how well the device is isolated from electrostatic fluctuations caused by unavoidable surface charges and other sources of electric noise. Current technology to fabricate quantum semiconductor devices relies on surface gates which impose strong limitations on the maximum distance from the surface where the confining electrostatic potentials can be engineered. Surface gates also introduce strain fields which cause imperfections in the semiconductor crystal structure. Another way to create confining electrostatic potentials inside semiconductors is by means of light and photosensitive dopants. Light can be structured in the form of perfectly parallel sheets of high and low intensity which can penetrate deep into a semiconductor and, importantly, light does not deteriorate the quality of the semiconductor crystal. In this work, we employ these important properties of structured light to form metastable states of photo-sensitive impurities inside a GaAs/AlGaAs quantum well structure in order to create persistent periodic electrostatic potentials at large predetermined distances from the sample surface. The amplitude of the light-induced potential is controlled by gradually increasing the light fluence at the sample surface and simultaneously measuring the amplitude of Weiss commensurability oscillations in the magnetoresistivity.

## 1. Introduction

Engineered micro- and nanostructures in semiconductors are—and will remain in the foreseeable future—the backbone of an overwhelming majority of microelectronic, optoelectronic, photonic and quantum devices. The underlying principle of all of the aforementioned devices is the creation of a system of confining energy barriers (or potentials) inside the host material in order to attain the desired quantum properties of charge carriers (electrons or holes) for specific applications.

Lateral superlattices (LSLs) are among the simplest structures that can be created in a semiconductor. These engineered structures are important for quantum physics and technology because they provide a new flexible platform for conducting rigorous and replicable testing of different models and approximations and, in the meantime, can be fabricated in a relatively simple fashion.

Semiconductor LSLs were first reported on high-index vicinal Si [1] and GaAs [2] surfaces. These LSLs were not of a high quality because of the excessive free-carrier scattering on the fluctuating vicinal steps. Later, LSLs were fabricated using holographic lithography combined with shallow etching to control the LSL’s modulation depth [3,4,5]. Nowadays, high-quality LSL is commonly fabricated using electron beam lithography combined with the gating or strain effect [6,7,8,9], and van der Waals stacking of zero-band-gap semiconducting graphene layers [10,11].

In the early 1990s, Weiss et al. demonstrated that one- and two-dimensional (1D and 2D, respectively) LSLs in AlGaAs/GaAs can be produced all optically using light interference patterns [12,13,14,15]. In this method, a spatially modulated photon flux with a 300–400 nm period selectively ionized deep Si-donors thus creating a periodic electrostatic potential near a quantum well (QW) containing a 2D electron gas (2DEG). It should be noted that free-space interferometers used in Refs. [12,13,14,15] are subject to acoustic noise and environmental fluctuations, limiting the illumination times to a few milliseconds. Another, much more robust technique to produce semiconductor LSLs by means of light-induced periodic electrostatic potentials was presented in our recent article where the light pattern to ionize the Si-donors was generated by a transmission phase diffraction grating (PDG) [16].

In the current work, we continue using the PDG technique to explore photoconductivity phenomena in GaAs/AlGaAs QW structures and provide evidence that there are at least two different mechanisms responsible for the creation of a sub-surface electrostatic potential using light depending on the illumination wavelength. Specifically, when the illumination wavelength lies in the near infrared, the potential is formed due to a direct photoionization process of deep-level impurities [16]. However, when visible light is used for the illumination, the mechanism involves a two-step process: (i) the generation of electron–hole pairs due to the strong inter-band light absorption and (ii) the subsequent capture of the photogenerated holes by the deep-level impurities. We also produce LSLs of higher quality compared to those reported in Ref. [16] and demonstrate how the LSL’s period can be changed by a factor of two by tuning the illumination wavelength while using one and the same PDG.

## 2. Materials and Methods

### 2.1. GaAs/AlGaAs QW-Structure

A GaAs/AlGaAs QW structure grown by molecular beam epitaxy was used in the experiments (Figure 1a). The QW was sandwiched between two δ-doping layers positioned 314 nm and 610 nm below the surface, whereas the center of the QW was 460 nm below the surface. The δ-doping layers were deposited in order to achieve an ultrahigh-mobility (≈1.1 × 10^7^ cm^2^/Vs) of the 2DEG confined in the GaAs layer [17]. A Hall bar was then fabricated by photolithography on the QW structure (Figure 1b). The width of the Hall bar was 60 μm and the separation between neighboring potential contacts on the Hall bar was 200 μm (Figure 1b).

The sample was placed in a helium-3 cryostat equipped with a split-coil superconducting solenoid with the magnetic field B directed perpendicular to the xy-plane. The magnetoresistivity $\rho_{xx}\left(B\right)=wV_{xx}/\left(L\;I\right)$ and the Hall resistivity $\rho_{xy}\left(B\right)=V_{xy}/I$ ($V_{xx}$ is the longitudinal voltage, $V_{xy}$ is the Hall voltage, w is the width of the Hall bar (i.e., 60 μm), L is the distance between neighboring potential contacts of the Hall bar (i.e., 200 μm) and I is the current passing through the Hall bar) were measured using a low-frequency lock-in amplifier.

### 2.2. Light Patterns Generated by PDGs

A periodic electrostatic potential in the Hall bar (Figure 1b) was induced by the interference pattern generated by a transmission PDG shown in Figure 2a. The PDG made of UV-grade fused silica was attached to the Hall bar (Figure 2b) with rubber glue (G in Figure 2c). Two optical fibers (OF in Figure 2c) with different single-mode operating wavelengths were used to deliver light to the sample (Figure 2c). Generally, if a linearly polarized light is launched into a single mode fiber, the exiting light will be a superposition of two orthogonally polarized HE_11_ modes, i.e., it will be an elliptically polarized light. Because the exact polarization state of the superposition was unknown in our experiments, a polarizer (Pol in Figure 2c) was placed in front of the light-turning prism (Pr in Figure 2c) to fix the light polarization at the PDG/Hall bar assembly for consistency.

The key element of the above sample-illumination system is a PDG and, in this connection, some relevant aspects of the PDG technique will be briefly presented below. We also note that the approach based on using structured light generated by a PDG to change the properties of glass is well known. Specifically, transmission PDGs have been used over the past thirty years for the inscription of fiber Bragg gratings by means of high-intensity laser radiation [18,19,20].

If light falls onto a PDG normally, the PDG will redirect it into several diffraction orders defined by the respective diffraction angles $\theta_{m}$ whose magnitudes in vacuum are given by $\sin^{-1}\left(m\lambda_{0}/\Lambda \right)$, where $\lambda_{0}$ is the wavelength of the incident light in vacuum, Λ is the PDG’s period and m is an integer satisfying the condition $\left|m\lambda_{0}/\Lambda \right|\leq 1$. In our experiments, $\left|m\right|\leq 1$, i.e., the PDG can generate only 0th and ±1 diffraction orders.

We first consider a situation when the power in the 0th diffraction order is much lower than that in the ±1 diffraction order. In this case, a two-beam interference pattern is formed by the diffracted light behind the PDG (Figure 3a) and, provided that the illumination light is quasi-monochromatic and spatially coherent, the two-beam interference pattern does not change with the distance from the PDG along the z-axis. The period of the interference pattern is given by $\lambda_{0}/\left[2\sin\left(\theta_{\pm 1}\right)\right]=\Lambda /2$ [21] both for vacuum and any transparent medium.

If a PDG generates a sufficiently strong 0th diffraction order in addition to ±1 diffraction orders, the interference pattern behind the BDG becomes more complex. The spatial structure of such a three-beam interference pattern periodically changes as the distance from the PDG and the observation point along the z-axis increases (Figure 3b). If the PDG is illuminated with a quasi-monochromatic and spatially coherent light, the intensity distribution in the xy-plane will be replicated when the distance is approximately changed by [22,23]:(1)$$\delta =\frac{\lambda_{0}}{1-{\left[1-{\left(\lambda_{0}/\Lambda \right)}^{2}\right]}^{\frac{1}{2}}}$$

Therefore, an interference pattern with a period of either Λ/2 or Λ can be seen in the xy-plane when the observation point is moved along the z-axis by approximately δ/4 (Figure 3b). These two distinct periods will be observed in both vacuum and any transparent material. Examples of experimentally obtained multi-beam interference patterns (i.e., Talbot patterns) can be found in Refs. [22,23,24].

As can be seen from Equation (1), the self-replication distance δ depends on both $\lambda_{0}$ and Λ. For a given Λ it will increase if the illumination wavelength $\lambda_{0}$ is decreased. This also implies that δ will be larger in an optically dense medium than in vacuum because the wavelength in a medium is inversely proportional to its refractive index.

The dependence of the self-replication distance of the interference pattern δ on the illumination wavelength allows one to double the period of the interference pattern at the sample provided that the distance D between the PDG and the sample surface remains fixed (Figure 4). For instance, D in Figure 4, it is such that for the wavelength $\lambda_{1}\left(\lambda_{1}<\lambda_{2}\right)$ an interference pattern with a period Λ/2 is produced at the sample surface (Figure 4a), whereas for the wavelength $\lambda_{2}$ the respective period is Λ (Figure 4b).

In our experiments, we illuminated the PDG at $\lambda_{1}$ = 637 nm and $\lambda_{2}$ = 780 nm (the FWHM spectral width of the laser sources is ≈0.5 nm) and clearly observed three-beam interference patterns produced by 0th and ±1 diffraction orders behind the PDG, as a substantial portion of the laser power (≈30% at 637 nm and ≈10% at 780 nm) was redirected into the respective 0th diffraction orders. As we will show below, LSLs with either a ≈700 nm period (i.e., ≈Λ/2) or a ≈1400 nm period (i.e., ≈Λ) were formed in the GaAs/AlGaAs QW-structure at $\lambda_{1}$ = 637 nm and $\lambda_{2}$ = 780 nm, respectively.

Finally, we would like to emphasize some key advantages of our all-optical PDG technique over the all-optical interferometric technique presented in Refs. [12,13,14,15]. The PDG technique (i) is based on a much more stable, monolithic optical setup (i.e., a single diffractive optical element bonded/glued to a semiconductor sample) featuring a simple initial optical alignment procedure and not requiring any subsequent realignment. Hence, prolonged and/or multistep sample illumination procedures, which can last for days, become possible. Moreover, the PDG technique (ii) utilizes a more compact optical setup (Figure 2b) which can be easily fit into a wide variety of small cryostats (e.g., helium-3 cryostats) using fiber-optics light delivery (Figure 2c); (iii) can use either monochromatic or polychromatic light; (iv) allows one, using one and the same PDG, to double the period of the light pattern by tuning the illumination wavelength.

## 3. Results

Two sets of experiments were performed with the Hall bar during which it was illuminated at $\lambda_{1}$ = 637 nm and $\lambda_{2}$ = 780 nm for different time durations and at different light intensities. The output laser power from the respective single-mode optical fibers, as well as the exposure times, were controlled electronically using two in-line fiber attenuators put in series to ensure a large dynamic range (i.e., 10 orders of magnitude) of possible light intensities at the sample surface. This was required to study the efficiency of photoionization processes involved in the creation of embedded LSLs. The sample temperature during the exposure to light was in the range of 260–280 mK.

In this work, we use the persistent photo-doping effect due to photoionization of deep-level impurities that are spatially separated from the QW. It is known that silicon (Si) as well as other impurities in Al*_x_*Ga_1−*x*_As can produce deep-level complexes, e.g., complexes containing a donor atom and a vacancy [25] which are often referred to as DX-centers [26]. Thermal ionization of DX centers requires temperatures above 100 K [25] and, therefore, at cryogenic temperatures below 77 K (i.e., the boiling point of liquid nitrogen) these centers are very stable.

The light intensities at the sample surface reported here were deduced from measurements performed with standard photodiode power meter sensors which provide an average value of the incident power. The same is true of the reported fluences F (fluence is the total optical energy delivered per unit area). We note, however, that the peak light intensity in the interference fringes generated by a 1D PDG differs from the average intensity in front of the PDG/Hall bar assembly. Nominally, the peak light intensity in a two-beam interference pattern produced by an ideal 1D PDG (i.e., 100% of the incident power is redirected only into ±1 diffraction orders) is two times higher than the intensity of the incident illumination light [21]. For a three-beam interference pattern produced by 0th and ±1 diffraction orders, this ratio is generally higher than one but also depends on the distance D between the PDG and the surface of the Hall bar due to the Talbot effect [22,23,24]. On the other hand, the peak intensity inside the sample is substantially reduced due to reflection losses at the sample surface (around 30% for both $\lambda_{1}$ and $\lambda_{2}$) and a non-perfect diffraction efficiency of the PDG used in the experiments. Therefore, the averaged intensities and fluences reported in this work are approximately equal to their peak values.

### 3.1. Hall Bar Illumination at $\lambda_{2}$
= 780 nm

The band gap of bulk Al*_x_*Ga_1-*x*_As at liquid-helium temperatures is approximately 1.52 eV, 1.67 eV and 1.82 eV for x = 0, x = 0.12 and x = 0.24, respectively [27], whereas the photon energy $E_{ph}$ of the near-infrared (NIR) light with $\lambda_{2}$ = 780 nm is 1.59 eV. Thus, the NIR light is very weakly attenuated by the 10 nm thick GaAs cap layer and the 35 nm thick GaAs QW but propagates without attenuation in the much thicker Al_0.12_Ga_0.88_As and Al_0.24_Ga_0.76_As barrier layers. Therefore, the NIR light can reach and ionize donors in the two δ-doping layers lying above and below the QW (Figure 1a and Figure 5). A periodically modulated flux of the NIR photons, realized by the interference of two (Figure 3a) or three (Figure 3b) plane waves at the surface of the Hall bar, will then photoionize the δ-doping donors (i.e., Si-donors [25,26]) and create two subsurface layers with a quasi-sinusoidal distribution of positive charges. We note that the weak absorption within the QW is not expected to play an important role in this process because the photo-generated electron-hole pairs quickly recombine and cannot change the charge state of Si-donors in the δ-doped regions. As a result, the 2DEG in the QW will move under the influence of a periodic electrostatic potential φ originating from these two layers with immobile positive charges. Following the previous discussion, the period of the subsurface charge distribution can be either Λ/2 or Λ, as indicated in Figure 5.

In this illumination series (Series 1), the light intensity at the sample surface was increased between consecutive illuminations but the exposure time was fixed at dt=10s for each illumination step. The magnetoresistivity $\rho_{xx}\left(B\right)$ of the Hall bar and the Hall resistivity $\rho_{xy}\left(B\right)$ as a function of applied magnetic field B were recorded after each illumination step and used to calculate the electron sheet concentration $n_{s}=B/\left[e\rho_{xy}\left(B\right)\right]$ and the Hall mobility $\mu_{H}=\rho_{xy}\left(B\right)/\left(B\rho_{0}\right)$. In the above expressions, e is the electron charge and $\rho_{0}$ is the zero-field resistivity.

Figure 6a,b show how the magnetoresistivity $\rho_{xx}\left(B\right)$ and the respective $n_{s}$ change with the accumulated fluence F. Commensurability Weiss oscillations are clearly seen in the $\rho_{xx}\left(B\right)$ traces within a certain range of accumulated fluence F: they are not present in trace 3 (F = 0.056 mJ/cm^2^), become well-developed in trace 4 (F = 0.21 mJ/cm^2^) and almost disappear in trace 9 (F = 1.98 mJ/cm^2^). Overall, Series 1 consisted of 30 separate illuminations at different intensities and lasted for 11 h. This demonstrates the remarkable stability of the PDG-based optical setup, which allows one to fine-tune light-induced electrostatic potentials during experiments.

In Figure 6b, the electron sheet concentration is plotted on a semi-logarithmic scale as a function of accumulated fluence F, which can be presented as $F=E_{ph}N_{ph}$, where $N_{ph}$ is the photon flux, i.e., the number of photons per unit area of the sample surface. The two distinct steps on this semi-logarithmic plot, which are approximated by linear functions f1=2.06+0.7F and $f2=1.75+6\times 10^{3}F$ which represent two different photoionization processes.

The first process (f1 in Figure 6b) corresponds to photoionization of deep donor centers in the δ-doped layers. According to f1, the generation efficiency of electrons can be written as $\delta n_{s}=0.7\delta F\approx 2\cdot 10^{-5}\delta N_{ph}$, i.e., $\approx 5\cdot 10^{4}$ photons produce one additional electron in the QW. This process clearly follows a linear dependence up to the saturation point occurring around F≈1mJ. Such linear dynamics are expected when only one type of dopants is involved in the photoionization. In our case, the dopants are localized in the thin δ-doping layers on both sides of the QW. More specifically, it is known that a certain portion of the Si-dopants form deep DX-centers whose presence leads to the persistent photo-conductivity effect [25,26]. The mechanism described above is somewhat similar to the one employed in novel photo-doped 2D materials [28,29]. In the latter case, charge carriers are also selectively generated in the photo-doped regions and then tunnel to the 2D conducting layer. The deep-donor ionization process described by f1 clearly leads to the creation of a periodic potential near the 2DEG, as evidenced by the emergence of Weiss oscillations in trace 4 as well as their presence in traces 5–9 until all deep donors in the δ-doping layers have been ionized at large fluences (Figure 6a).

For the other process ($f_{2}$ in Figure 6b), the photon-electron conversion efficiency is approximately four orders of magnitude higher than in the first process, i.e., it is on the order of ${2\cdot 10}^{-1}$. It can be seen that this photoionization process cannot be fully described by the linear function f2, which is likely due to the presence of different types of dopants or states, for example, various surface states and background impurities. It should be emphasized that these donor states/impurities are randomly distributed over the surface and in the volume of the barrier layers shown in Figure 1a and, therefore, their ionization by the structured light does not result in the formation of a well-defined lateral potential. Indeed, there is no evidence of Weiss oscillations in traces 2 and 3 (Figure 6a) even though they were recorded after the photoionization of background impurities had been completed (see the corresponding data points in Figure 6b).

### 3.2. Hall Bar Illumination at $\lambda_{1}$
= 637 nm

In this illumination series (Series 2), the visible (VIS) radiation with $\lambda_{1}$ = 637 nm is strongly absorbed in the sample as its photon energy $E_{ph}$ is considerably larger than the AlGaAs’s band gap at liquid-helium temperatures. According to the absorption coefficient data in bulk Al*_x_*Ga_1-*x*_As measured at liquid-helium temperatures for different compositions x [27,30] we estimate that ≈60% and ≈30% of the incident photon flux reaches the top and the bottom δ-doping layer, respectively, whereas the rest of the VIS light is absorbed in the AlGaAs barrier layers.

Series 2 consisted of two separate experiments in which the exposure time dt was 10 s as in Series 1 (blue data points in Figure 7b) and 1 s (red data points in Figure 7b). The light intensity in these two experiments gradually increased between consecutive illuminations. The magnetoresistivity $\rho_{xx}\left(B\right)$-traces for different F and $n_{s}$-values as a function of F are presented in Figure 7a,b, respectively.

When the exposure time dt in Series 2 was 10 s, the increase in electron concentration $n_{s}$ was very fast due to very strong interband absorption of light with $\lambda_{1}$ = 637 nm (blue squares in Figure 7b) and only one trace exhibiting Weiss oscillations was captured. In order to carefully examine the evolution of $n_{s}$ as a function of F, we reduced the increments in the accumulated F by decreasing dt to 1 s (red symbols in Figure 7b). As a result, the duration of the experiment became limited by the helium-3 cryostat hold time and magnetoresistivity traces with only one magnetic field direction were recorded.

Once again, one can see that the observed growth of $n_{s}$ as a function of F cannot be described over the full interval by a linear function, i.e., $f3=1.685+3.2\times 10^{3}F$, which implies that a more complex mechanism governs this process—it is not just a strong interband absorption with one type of impurities involved [31,32]. One possible mechanism may be described as follows: (i) the VIS light is strongly absorbed in the AlGaAs barrier layers and produces electron-hole pairs near the δ-doping layers containing DX centers; (ii) some of the photo-generated holes are captured by DX centers, change their charge states and consequently create a periodic distribution of immobile positive charges in the δ-doping layers with a periodic spatial distribution that follows the light pattern; (iii) the unpaired free electrons drift towards and are eventually captured by the QW increasing $n_{s}$ of the 2DEG. Evidently, this new mechanism is very efficient (i.e., $\delta n_{s}=3.2\times 10^{3}\delta F\approx 10^{-1}\delta N_{ph}$) and, in principle, can be employed in photon-to-spin devices if some photo-active layers are introduced [33,34] at the tunneling distance from the quantum-dot structure designed to store and process quantum information [35,36].

### 3.3. Wavelength Dependence of LSLs’ Periods

After processing the $\rho_{xx}\left(B\right)$-traces from Series 1 and 2 according to the procedure described in Ref. [16], the periods (i.e., a) and amplitudes of the respective light-induced electrostatic potentials can be obtained with high accuracy.

For $\lambda_{2}$ = 780 nm, we find a= 1420 nm (i.e., a≈Λ), while for $\lambda_{1}$ = 637 nm, a= 710 nm (i.e., a≈Λ/2). These results indicate that the distance D between the PDG and the Hall bar was such that the periods of the multi-beam interference patterns at the sample surface were, respectively Λ/2 and Λ for $\lambda_{1}$ = 637 nm and $\lambda_{2}$ = 780 nm, as it is shown in Figure 4. Had the distance D happened to be ≈1 μm larger or smaller, the situation with the LSLs’ periods would have been reversed.

The distance D in our experiments was approximately equal to the diameter of the bonding wire (W in Figure 2c) that is ≈50 μm. Because we used quasi-monochromatic light sources (i.e., FWHM spectral width ≈ 0.5 nm) in the above experiments, well-defined three-beam interference patterns were generated hundreds of micrometers away from the PDG. In this respect, the production of LSLs with Λ/2 and Λ period at the respective wavelengths we consider to be a coincidence. However, this uncertainty can be removed by using superluminescent VIS and NIR light sources with the FWHM spectral width ≈ 20 nm. In this case, the three-beam interference pattern (Figure 3b and Figure 4) is expected to transform into a two-beam interference pattern with a stable period of Λ/2 for D>100 μm, thus making the LSL’s period well defined and fully predictable. As a consequence, the ability to change the LSL’s period by tuning the illumination wavelength disappears in this case.

Finally, following the formalism presented in Ref. [16], we determine the dimensionless LSL’s modulation potential $\eta ={eV}_{0}/E_{F}$, where $V_{0}$ is the modulation amplitude of the light-induced electrostatic potential and $E_{F}$ is the Fermi energy of the 2DEG. The dimensionless potential η for the NIR and VIS light is 2.2% and 0.6%, respectively, whereas the Fermi energy for the $\rho_{xx}\left(B\right)$-traces in Figure 8a,b is 8.71 meV and 7.72 meV, respectively. Consequently, $V_{0}$ for the NIR and VIS light is 0.19 mV and 0.046 mV, respectively. The above values for $V_{0}$ are close to those reported in Refs. [14,15].

## 4. Discussion

Our experimental data demonstrate that depending on the wavelength, there are two different mechanisms by means of which structured light can induce periodic electrostatic potentials near a high-mobility 2DEG.

The first mechanism is based on direct photoionization of deep-donor levels in the δ-doping layers when the photon energy is smaller than the band gap of the host semiconductor structure. In this case, the QW with a 2DEG and the δ-doping layers can be positioned very deep (>1 μm) below the sample surface as the light absorption in the semiconducting layers surrounding the QW is very weak. The quality of the resultant quantum structures is expected to be high as the extent to which they will be influenced by surface charges and strain fields originating from surface gates and other surface imperfections is now dramatically reduced.

The second mechanism takes place when the photon energy is larger than the band gap of the host semiconductor structure. This mechanism is based on changing the charge state of donors in the δ-doping layers by holes generated in the volume. In this case the maximum depth at which the QW and the δ-doping layers can be positioned is determined by the absorption coefficient of light at the wavelength used. Taking into account that interband absorption coefficients for AlGaAs lie in the range from 1× 10^4^ cm^−1^ to 2 × 10^4^ cm^−1^ [27] depending on the difference between the photon energy and the band gap, quantum structures can be created at depths limited to ≈0.5 μm.

Despite the advantages offered by the all-optical approach, it has not found wide acceptance so far. We see two reasons for that. First, free-space interferometers are proposed in Refs. [12,13,14,15] for this task is difficult to operate inside a cryostat at low temperatures due to their mechanical instability. Second, the minimum period of the resultant LSL is determined by the diffraction limit of the illumination light (i.e., 300–700 nm). Because the light-induced electrostatic potential can appreciably affect the behavior of the 2DEG when its period is smaller than the quantum mean free path of electrons in the 2DEG, only high-mobility samples can be used in such studies.

The novel PDG technique essentially removes the first constraint from the all-optical approach due to its simplicity and the remarkable stability of the pertinent optical setup, opening up the possibility of continuous experiments on imprinting quantum structures at cryogenic temperatures using multiple illuminations.

The second constraint is removed due to the increasing availability of semiconductor heterostructures whose 2DEG mobilities are ≈10^7^ cm^2^/Vs [17]. The quantum mean free path of charge carriers in such samples exceeds several micrometers [37,38,39,40], implying that large-period light-induced LSLs become suitable for experiments both at low magnetic fields and in the fractional quantum Hall effect regime [7,15].

To conclude, the all-optical technique allows one to create sub-surface periodic potentials near a QW in a wide variety of semiconductors if regions containing deep-level impurities are introduced at certain distances from the QW. It is also worth mentioning that our experiments, as well as the experiments by Weiss et al. [12,13,14,15], were conducted on non-planar Hall bar mesa-structure devices prepared by standard lithography. Importantly, because the height/depth of the non-planar surface structures was very small compared to the wavelength of the illumination light, they did not have any noticeable effect on the interference pattern inside the semiconductor samples. During the current and previous experiments, the optical technique was applied to devices that had fully opaque Ohmic contact areas. This did not affect the measurements thanks to using the standard four-probe technique. Moreover, surface electrodes can be made of transparent materials such as standard indium-tin-oxide (ITO) conducting films. We believe that the simplicity, robustness and cost effectiveness of the PDG technique will allow its adoption and broad use in laboratories that are pursuing the realization of quantum circuits on the semiconductor platforms.

## Figures and Tables

**Figure 1 nanomaterials-13-01622-f001:**
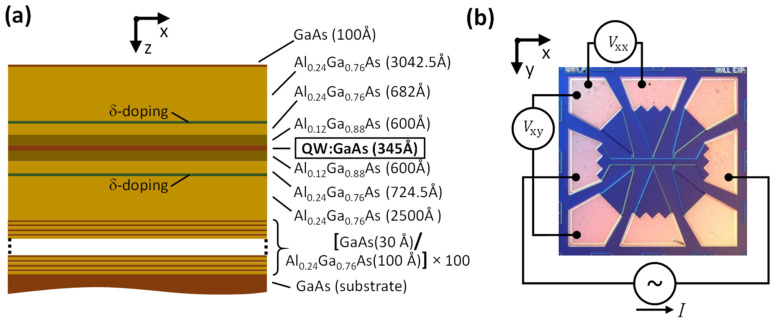
Sample and experiment. (**a**) The structure of the GaAs/AlGaAs wafer; “δ-doping” denotes the Si-doping layers. (**b**) A differential interference contrast microscopy image of the 60 μm-wide Hall bar fabricated on the GaAs/AlGaAs QW structure shown in (**a**).$V_{xx}$, $V_{xy},$ and I are defined in the text.

**Figure 2 nanomaterials-13-01622-f002:**
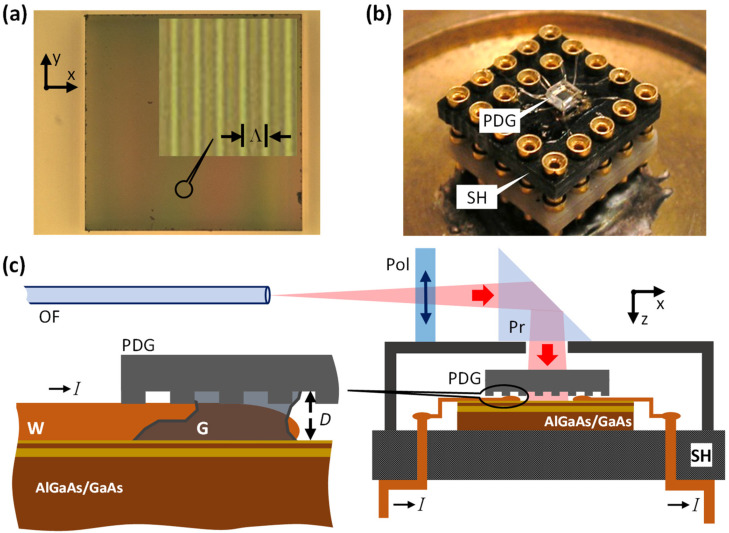
The PDG technique to imprint LSLs in semiconductors. (**a**) An optical microscopy image of the ≈1 × 1 mm^2^ PDG used in the experiments. The period of the PDG is denoted by Λ (Λ = 1400 nm). (**b**) An image of the PDG mounted on the 12 × 12 mm^2^ sample holder (SH). (**c**) Schematic of the optical setup used to deliver the illumination light to the Hall bar. In (**c**), OF is the single-mode optical fiber, W is the bonding wire (≈50 μm in diameter), D is the distance between the PDG and the surface of the Hall bar, G is the rubber glue, Pol is the polarizer, Pr is the light-turning prism and I is the current through the Hall bar.

**Figure 3 nanomaterials-13-01622-f003:**
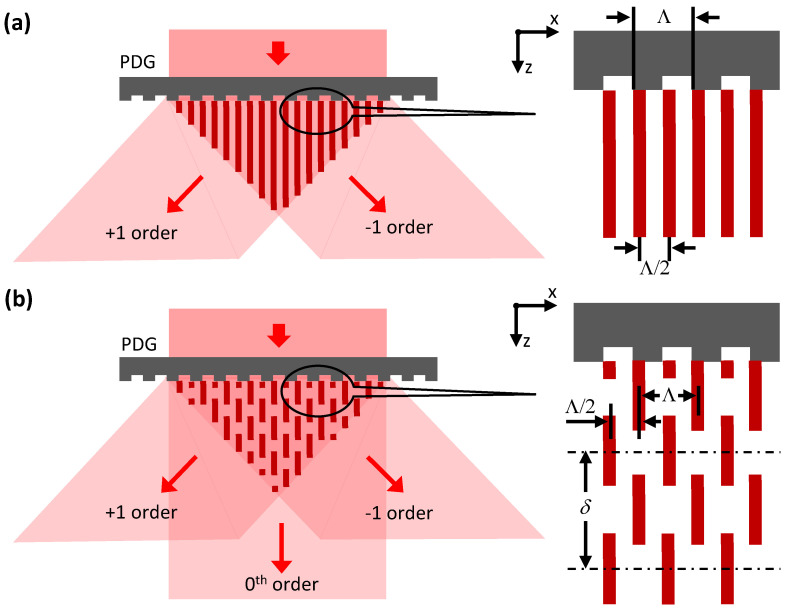
Interference patterns formed by a 1D PDGs. (**a**) A PDG with a period Λ generates ±1 diffraction orders; the period of the resultant interference pattern in the xy-plane (i.e., Λ/2) does not change with the distance from the PDG to the observation point along the z-axis. (**b**) A PDG with a period Λ generates 0th and ±1 diffraction orders; the period of the interference pattern in the xy-plane can be either Λ/2 or Λ depending on the z-coordinate. The self-replication of the interference pattern occurs when the observation point is moved along the z-axis by a distance δ.

**Figure 4 nanomaterials-13-01622-f004:**
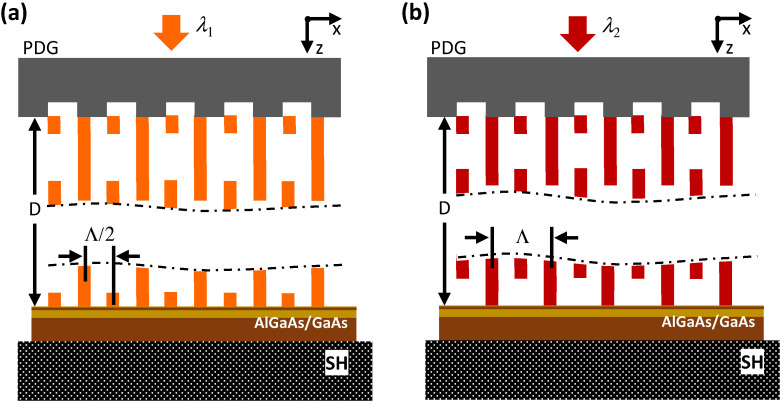
Doubling the period of the interference pattern produced by a 1D PDG at the sample surface by tuning the wavelength of illumination light. (**a**) The self-replication distance of the interference pattern $\delta_{1}$ at the wavelength $\lambda_{1}\left(\lambda_{1}<\lambda_{2}\right)$ is such that an interference pattern with a period Λ/2 is produced at the sample surface. (**b**) The self-replication distance of the interference pattern $\delta_{2}$ at the wavelength $\lambda_{2}$ is such that an interference pattern with a period Λ is produced at the sample surface. One and the same PDG with a period Λ is depicted in both (**a**) and (**b**). The distance D between the PDG and the surface of the Hall bar is the same in (**a**,**b**).

**Figure 5 nanomaterials-13-01622-f005:**
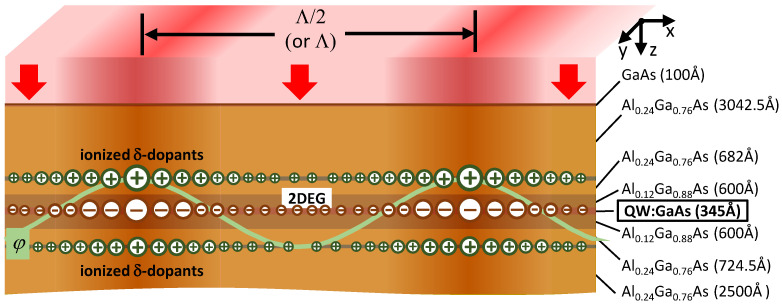
Using the interference pattern produced by a 1D PDG to generate a periodic electrostatic potential in δ-doped GaAs/AlGaAs QW-structures. δ-dopants ionized by light and electrons of the 2DEG in the QW are denoted by “+” and “−“, respectively. Larger sizes of the “+” and “−“ signs correspond to regions with higher concentrations of the ionized δ-dopants and electrons. The light-induced electrostatic potential φ, whose period along the x-axis can be either Λ/2 or Λ depending on the distance D between the PDG and the surface of the Hall bar, is represented by the light-green sinusoid.

**Figure 6 nanomaterials-13-01622-f006:**
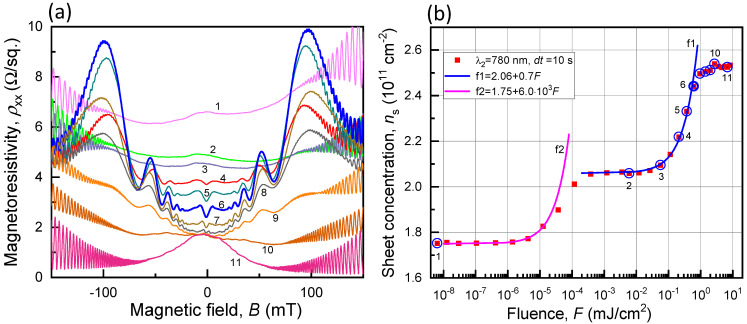
Changes in the magnetoresistivity and electron concentration of the Hall bar illuminated through the PDG using the NIR light with $\lambda_{2}$ = 780 nm. (**a**) Traces of magnetoresistivity $\rho_{xx}\left(B\right)$ obtained at different accumulated fluences (see (**b**)). Traces marked with larger numbers correspond to higher accumulated fluences. (**b**) Sheet concentration $n_{s}$ as a function of accumulated fluence F. The curves in magenta (f2) and blue (f1) represent linear fits for the two different electron-generation mechanisms described in the text. The trace numbering in (**a**) corresponds to the data point numbering in (**b**), with each number representing a certain accumulated fluence. The trace marked by 6 in (**a**) was chosen for the fitting procedure described below (for details see text).

**Figure 7 nanomaterials-13-01622-f007:**
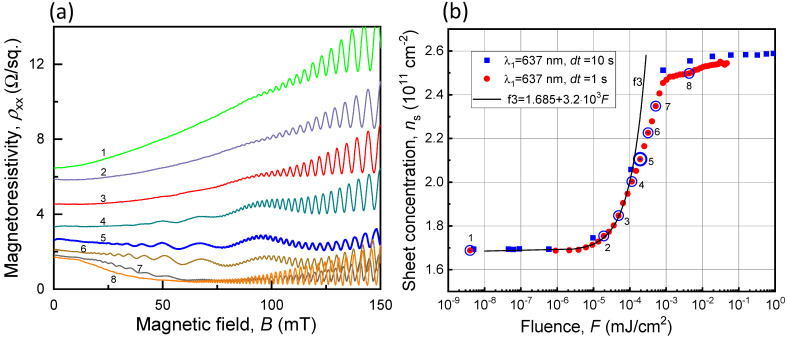
Changes in the magnetoresistivity and electron concentration of the Hall bar illuminated through the PDG using the VIS light with $\lambda_{1}$ = 637 nm. (**a**) Traces of magnetoresistivity $\rho_{xx}\left(B\right)$ obtained at different accumulated fluences (see (**b**)). Traces marked with larger numbers correspond to higher accumulated fluences. (**b**) Sheet concentration $n_{s}$ as a function of accumulated fluence F. The curve in black (f3) represents a linear fit for the respective electron-generation mechanism described in the text. The trace numbering in (**a**) corresponds to the data point numbering in (**b**), with each number representing a certain accumulated fluence. The trace marked by 5 in (**a**) was chosen for the fitting procedure in Figure 8b.

**Figure 8 nanomaterials-13-01622-f008:**
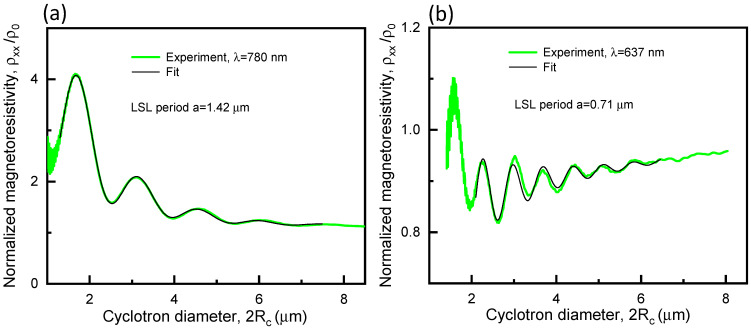
Fitting of the normalized magnetoresistivity $\rho_{xx}/\rho_{0}$ ($\rho_{0}$ is the zero-field resistivity) as a function of the cyclotron diameter $2R_{c}$ ($2R_{c}=\hbar \sqrt{8\pi n_{s}}/\left(eB\right)$, where ℏ is the reduced Planck constant). (**a**) Fitting of trace 6 in Figure 6a; $n_{s}$ = 2.44 × 10^11^ cm^−2^ according to the data point 6 in Figure 6b. (**b**) Fitting of trace 5 in Figure 7a; $n_{s}$ = 2.16 × 10^11^ cm^−2^ according to the data point 5 in Figure 7b. The fitting curves to deduce the LSL’s period a were obtained using the formalism from Ref. [16]. The PDG’s period Λ is equal to 1400 nm.

## Data Availability

The data that support the findings of this study are available from the corresponding author upon reasonable request.

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
