# Peer review of "An Optical Technique to Produce Embedded Quantum Structures in Semiconductors"

_nanomaterials, 2023, doi:10.3390/nano13101622_

Round 1

Reviewer 1 Report

This is an interesting and well-written report that describes the use of phase diffraction gratings illuminated with different wavelengths of light to selectively ionize Si dopant atoms and modify the electrostatic potential adjacent to a 2DEG. The authors should consider the points below to strengthen their manuscript.

1.     Is there any temperature dependence of the diffraction pattern created due to thermal expansion of the PDG?

2.     I assume this needs to be done so cold to prevent thermal ionization of all of the Si atoms in the delta-doped layers? If this is correct, for the benefit of the reader perhaps this point should be made somewhere, along with comparison between the donor energy of Si in Al0.24Ga0.76As and kT at 280 mK.

3.     Why not plot Figs 6a and 7a using the same x-axis (i.e. one or both field directions)?

4.     Why did ns increase so much faster for 637 nm illumination compared with 780 nm illumination?

5.     Lines 375-379: Using superluminescent VIS and NIR light sources would transform PDG output to two-beam interference pattern. I can see the benefit of this, but wouldn’t this also remove the ability to double the LSL period with wavelength (as per lines 188-189) i.e., the experiment done in this paper?

Author Response

Manuscript ID: nanomaterials-2341843

Response to the Reviewer 1 comments.

We thank the Reviewer for his or her positive feedback and insightful comments on the manuscript. Below, we provide answers to the Reviewer’s questions.

  1. Is there any temperature dependence of the diffraction pattern created due to thermal expansion of the PDG?

Our Response:

The PDG that was used in our experiments was made of UV grade fused silica. This material has a very low thermal expansion coefficient within a wide temperature range, specifically, it is ≈ 0.5 x 10-6/K at 300 K, -0.8 x 10-6/K (negative) at 50 K, and 0 at 0 K [M. Okaji, N. Yamada, K. Nara, and H. Kato, “Laser interferometric dilatometer at low temperatures: application to fused silica SRM 739,” Cryogenics 35(12), 887–891 (1995)]. Hence, the average change in the PDG’s period when the PDG is cooled down from 300 K to 0 K will only be 0.25 x 10-6/K. The respective change in the diffraction angles and the resultant interference pattern is therefore totally negligible.

The following text was added on p. 3, line 98:

The PDG made of UV-grade fused silica was…. 

  1. I assume this needs to be done so cold to prevent thermal ionization of all of the Si atoms in the delta-doped layers? If this is correct, for the benefit of the reader perhaps this point should be made somewhere, along with comparison between the donor energy of Si in Al0.24Ga0.76As and kT at 280 mK.

Our Response:

Silicon as well as other impurities in AlxGa1-xAs are known to produce both shallow-level and deep-level complexes, e.g., complexes containing a donor atom and a vacancy. The latter are often referred to as DX-centers. This is why a red-light illumination is commonly used to activate all impurities and thus increase carrier concentration and mobility. The ionization energy of DX-centers is very large compared to shallow hydrogen-like donors and requires temperatures above 100 K to thermally ionize them. Therefore, at cryogenic temperatures below 77 K (i.e., the boiling point of liquid nitrogen) these centers are very stable, which leads to a long-lasting photo-doping effect that was used in this work.

The following text was added on p.6, line 207:

In this work we use the persistent photo-doping effect due to photoionization of deep-level impurities that are spatially separated from the QW. It is known that silicon as well as other impurities in AlxGa1-xAs can produce deep-level complexes, e.g., complexes containing a donor atom and a vacancy [25] which are often referred to as DX-centers [26]. Thermal ionization of DX-centers requires temperatures above 100 K [25] and, therefore, at cryogenic temperatures below 77 K (i.e., the boiling point of liquid nitrogen) these centers are very stable.

  1. Why not plot Figs 6a and 7a using the same x-axis (i.e. one or both field directions)?

Our Response:

In the experiment presented in Fig. 7a, due to the strong interband absorption at 637 nm, we had to record tens of traces at slightly increasing fluences in order to identify the optimum conditions for the formation of the LSL potential. The duration of the experiments at 637 nm was simply limited by the He3 cryostat hold time which allowed us to record traces for only one magnetic field direction. We knew from the experiments at 780 nm (Fig. 6a) that the traces were symmetric with respect to the magnetic field.

The following text was added on p. 10, line 350:

As a result, the duration of the experiment became limited by the helium-3 cryostat hold time and magnetoresistivity traces with only one magnetic field direction were recorded.

  1. Why did nsincrease so much faster for 637 nm illumination compared with 780 nm illumination?

Our Response:

As discussed in the text on. p.9, the photon energy of the red light is larger than the band gap of the semiconductor. Therefore, the red light encounters a very efficient inter-band absorption, much stronger than the infrared light at  = 780 nm.

The following sentence was added to the text on p.10, line 346:

…the increase in electron concentration  was very fast due to a very strong interband absorption of light with  = 637 nm (blue squares in Figure 7b)…

  1. Lines 375-379: Using superluminescent VIS and NIR light sources would transform PDG output to two-beam interference pattern. I can see the benefit of this, but wouldn’t this also remove the ability to double the LSL period with wavelength (as per lines 188-189) i.e., the experiment done in this paper?

Our Response:

This is correct. Here as well as in our previous work (i.e., Ref. 15 in the Manuscript), the distance between the PDG and the sample could not be accurately controlled and, as a result, the period of the three-beam interference pattern at the sample surface was unpredictable. It is true that by tuning the illumination wavelength the period can be made either lambda/2 or lambda, but this complicates the experiment. In practical applications it is easier to work with the known smaller period (i.e., lambda/2), which can be achieved by placing the sample farther from the PDG and using a broadband light source.

The following text was added on p.11, line 395:

As a consequence, the ability to change the LSL’s period by tuning the illumination wavelength disappears in this case.

Reviewer 2

We concur that the manuscript could be improved by emphasizing its novelty aspect and are thankful to the Reviewer for pinpointing this shortcoming. However, we respectfully disagree with the Reviewer regarding their doubts that the proposed technique may not have practical applications. To address the Reviewer’s concerns, we have amended the text, as it will be shown below.

  1. The major conclusion in this manuscript (nanomaterials‐2341843), the device structure, major experiment procedure are similar with those in an article by the same author with the title of Inscription of lateral superlattices in semiconductors using structured light” in J. Appl. Phys. 132, 044301 (2022). Though the authors cited that JAP paper, the new findings in the manuscript are not clear. Can the authors clarify the new significant finding of this manuscript (nanomaterials‐2341843)?

Our Response:

The novelty of the current work is stated concisely but clearly in the Introduction. It is: i) An evidence that there are at least two different mechanisms responsible for the creation of a sub-surface electrostatic potential using light (not shown in Ref. [15]); ii) a significantly higher quality of the light-induced electrostatic potential compared to our previous results (i.e., Ref. [15]); and iii) experimental demonstration that the superlattice period can be changed by tuning the illumination wavelength (in principle, can be deduced from Ref. [15] but was not demonstrated in Ref. [15]). Among these new results, the main one is the demonstration of a new mechanism of light-semiconductor interaction that leads to the creation of a periodic subsurface electrostatic potential (i.e., a superlattice). We did not elaborate on this in the Introduction, but a whole section was dedicated to the new mechanism in Section 3.2 (Hall bar illumination at  = 637 nm). We agree, that we didn’t explicitly emphasize the “novelty” in the Section 3.2 – it is just “Hall bar illumination at  = 637 nm.”

Comparing the two illumination series at  = 780 nm and  = 637 nm, we can clearly identify two different mechanisms responsible for the creation of an embedded electrostatic potential. Whereas the first one, at  = 780 nm (reported in Ref. [15]), is based on a direct photo-ionization process of deep-level centers, the other mechanism, at  = 637 nm, involves a two-step process (not reported in Ref. [15]). At the first step the light encounters strong inter-band absorption generating electron-hole pairs. The generated electrons and holes have certain probabilities to be recombine or be captured by deep level traps. Due to the superior quality of the material, the concentration of the background traps is very low and the whole process is dominated by intentionally introduced dopants during the growth process. In the presented experimental data, it is clear that the photo-generated holes are captured by the neutral centers and the unpaired electrons diffuse and eventually end up in the quantum well leading to an increase in the 2DEG sheet concentration, as shown in Fig. 7b. The hole capture process is very efficient, which is evident from the dependence of the photo-induced carrier concentration on fluence of the carriers. In the VIS illumination series, the optimal LSL potential is created at F=2∙10-4 mJ/cm2, while in the NIR series the maximum modulation is reached at a 3∙103 times larger fluence, i.e., F=0.6 mJ/cm2. It is clear that the VIS-light mechanism has a very high efficiency, which can also be employed in quantum devices that interface photon and spin.

We also note, that the proposed technique is new and, in this respect, we consider any improvements to the technique (i.e., the higher quality of the LSL created and the wavelength tunability of the LSL’s period) to be important because we do believe that this technique has a great practical value.

The following text has been added to Introduction on p.2, line 62:

…using light depending on the illumination wavelength. Specifically, when the illumination wavelength lies in the near-infrared (NIR), the potential is formed due to a direct photo-ionization process of deep-level impurities [15]. However, when visible (VIZ) light is used for the illumination, the mechanism involves a two-step process: i) the generation of electron-hole pairs due to the strong inter-band light absorption and ii) the subsequent capture of the photo-generated holes by the deep-level impurities.

The following minor correction was introduced to a sentence in Section 3.2 (p. 10, line 362):

Evidently, this new mechanism is very efficient …

  1. Though structured light proposed in this manuscript may have some potential advantages, considering the different composition in different semiconductor layer (different bandgap to cause different absorption), different doping (caused different free carrier absorption), furthermore, most of the real semiconductor device is not planar, the light may not generate uniform pattern inside semiconductor. In addition, for most of the semiconductor devices, metal electrode is needed to be on top of semiconductor to inject carriers (above the active region). How can the light penetrate the metal and get inside semiconductor? Therefore, the Reviewer doubt this technique to have any practical application.

Our Response is subdivided into three subsections:

(2.1) Though structured light proposed in this manuscript may have some potential advantages, considering the different composition in different semiconductor layer (different bandgap to cause different absorption), different doping (caused different free carrier absorption).

The all-optical technique allows one to create sub-surface periodic potentials near a QW in a wide variety of semiconductors. To do so, deep-level impurities need to be introduced at a certain distance from the QW.

(2.2)

most of the real semiconductor device is not planar, the light may not generate uniform pattern inside semiconductor layer…”

Our experiments, as well as the experiments by Weiss et al, were conducted on a non-planar Hall bar mesa-structure devices prepared by standard lithography, as is described in the manuscript. The height/depth of the non-planar features is very small compared to the wavelength of the illumination light and therefore does not have any noticeable effect on the phenomena under investigation.

(2.3)

“…for most of the semiconductor devices, metal electrode is needed to be on top of semiconductor to inject carriers (above the active region). How can the light penetrate the metal and get inside semiconductor?”

This issue of opaque metal electrodes can be tackled in a few different ways. To start with, our experiments, as well as the experiments by Weiss et al, were carried out with devices that had fully opaque Ohmic contact areas. This did not affect the measurements thanks to using the standard four-probe technique. In other words, there are several experimental techniques that allow one to minimize the contribution of non-illuminated areas. Additionally, surface electrodes can be made from transparent materials, such as standard indium-tin-oxide (ITO) conducting films.

Finally, the title of this manuscript is “An optical technique to produce embedded quantum structures in semiconductors,” which we have successfully demonstrated. A device-level discussion lies outside the scope of this manuscript but will be the subject of our further publications.

To emphasize potential practical implications, we have added the following text on p. 12:

To conclude, the all-optical technique allows one to create sub-surface periodic potentials near a QW in a wide variety of semiconductors if regions containing deep-level impurities are introduced at certain distances from the QW. It is also worth mentioning that our experiments, as well as the experiments by Weiss et al [12-15], were conducted on non-planar Hall bar mesa-structure devices prepared by standard lithography. Importantly, because the height/depth of the non-planar surface structures was very small compared to the wavelength of the illumination light, they did not have any noticeable effect on the interference pattern inside the semiconductor samples. During the current and previous experiments, the technique was applied to devices that had fully opaque Ohmic contact areas. This did not affect the measurements thanks to using the standard four-probe technique. Moreover, surface electrodes can be made of transparent materials, such as standard indium-tin-oxide (ITO) conducting films.

On behalf of all Authors

Sergei Studenikin and Cyril Hnatovsky                                 April 30, 2023

Reviewer 2 Report

Please refer to the attached review report.

Author Response

Manuscript ID: nanomaterials-2341843

Response to the Reviewer 2 comments.

We concur that the manuscript could be improved by emphasizing its novelty aspect and are thankful to the Reviewer for pinpointing this shortcoming. However, we respectfully disagree with the Reviewer regarding their doubts that the proposed technique may not have practical applications. To address the Reviewer’s concerns, we have amended the text, as it will be shown below.

  1. The major conclusion in this manuscript (nanomaterials‐2341843), the device structure, major experiment procedure are similar with those in an article by the same author with the title of Inscription of lateral superlattices in semiconductors using structured light” in J. Appl. Phys. 132, 044301 (2022). Though the authors cited that JAP paper, the new findings in the manuscript are not clear. Can the authors clarify the new significant finding of this manuscript (nanomaterials‐2341843)?

Our Response:

The novelty of the current work is stated concisely but clearly in the Introduction. It is: i) An evidence that there are at least two different mechanisms responsible for the creation of a sub-surface electrostatic potential using light (not shown in Ref. [15]); ii) a significantly higher quality of the light-induced electrostatic potential compared to our previous results (i.e., Ref. [15]); and iii) experimental demonstration that the superlattice period can be changed by tuning the illumination wavelength (in principle, can be deduced from Ref. [15] but was not demonstrated in Ref. [15]). Among these new results, the main one is the demonstration of a new mechanism of light-semiconductor interaction that leads to the creation of a periodic subsurface electrostatic potential (i.e., a superlattice). We did not elaborate on this in the Introduction, but a whole section was dedicated to the new mechanism in Section 3.2 (Hall bar illumination at  = 637 nm). We agree, that we didn’t explicitly emphasize the “novelty” in the Section 3.2 – it is just “Hall bar illumination at  = 637 nm.”

Comparing the two illumination series at  = 780 nm and  = 637 nm, we can clearly identify two different mechanisms responsible for the creation of an embedded electrostatic potential. Whereas the first one, at  = 780 nm (reported in Ref. [15]), is based on a direct photo-ionization process of deep-level centers, the other mechanism, at  = 637 nm, involves a two-step process (not reported in Ref. [15]). At the first step the light encounters strong inter-band absorption generating electron-hole pairs. The generated electrons and holes have certain probabilities to be recombine or be captured by deep level traps. Due to the superior quality of the material, the concentration of the background traps is very low and the whole process is dominated by intentionally introduced dopants during the growth process. In the presented experimental data, it is clear that the photo-generated holes are captured by the neutral centers and the unpaired electrons diffuse and eventually end up in the quantum well leading to an increase in the 2DEG sheet concentration, as shown in Fig. 7b. The hole capture process is very efficient, which is evident from the dependence of the photo-induced carrier concentration on fluence of the carriers. In the VIS illumination series, the optimal LSL potential is created at F=2∙10-4 mJ/cm2, while in the NIR series the maximum modulation is reached at a 3∙103 times larger fluence, i.e., F=0.6 mJ/cm2. It is clear that the VIS-light mechanism has a very high efficiency, which can also be employed in quantum devices that interface photon and spin.

We also note, that the proposed technique is new and, in this respect, we consider any improvements to the technique (i.e., the higher quality of the LSL created and the wavelength tunability of the LSL’s period) to be important because we do believe that this technique has a great practical value.

The following text has been added to Introduction on p.2, line 62:

…using light depending on the illumination wavelength. Specifically, when the illumination wavelength lies in the near-infrared (NIR), the potential is formed due to a direct photo-ionization process of deep-level impurities [15]. However, when visible (VIZ) light is used for the illumination, the mechanism involves a two-step process: i) the generation of electron-hole pairs due to the strong inter-band light absorption and ii) the subsequent capture of the photo-generated holes by the deep-level impurities.

The following minor correction was introduced to a sentence in Section 3.2 (p. 10, line 362):

Evidently, this new mechanism is very efficient …

  1. Though structured light proposed in this manuscript may have some potential advantages, considering the different composition in different semiconductor layer (different bandgap to cause different absorption), different doping (caused different free carrier absorption), furthermore, most of the real semiconductor device is not planar, the light may not generate uniform pattern inside semiconductor. In addition, for most of the semiconductor devices, metal electrode is needed to be on top of semiconductor to inject carriers (above the active region). How can the light penetrate the metal and get inside semiconductor? Therefore, the Reviewer doubt this technique to have any practical application.

Our Response is subdivided into three subsections:

(2.1) Though structured light proposed in this manuscript may have some potential advantages, considering the different composition in different semiconductor layer (different bandgap to cause different absorption), different doping (caused different free carrier absorption).

The all-optical technique allows one to create sub-surface periodic potentials near a QW in a wide variety of semiconductors. To do so, deep-level impurities need to be introduced at a certain distance from the QW.

(2.2)

most of the real semiconductor device is not planar, the light may not generate uniform pattern inside semiconductor layer…”

Our experiments, as well as the experiments by Weiss et al, were conducted on a non-planar Hall bar mesa-structure devices prepared by standard lithography, as is described in the manuscript. The height/depth of the non-planar features is very small compared to the wavelength of the illumination light and therefore does not have any noticeable effect on the phenomena under investigation.

(2.3)

“…for most of the semiconductor devices, metal electrode is needed to be on top of semiconductor to inject carriers (above the active region). How can the light penetrate the metal and get inside semiconductor?”

This issue of opaque metal electrodes can be tackled in a few different ways. To start with, our experiments, as well as the experiments by Weiss et al, were carried out with devices that had fully opaque Ohmic contact areas. This did not affect the measurements thanks to using the standard four-probe technique. In other words, there are several experimental techniques that allow one to minimize the contribution of non-illuminated areas. Additionally, surface electrodes can be made from transparent materials, such as standard indium-tin-oxide (ITO) conducting films.

Finally, the title of this manuscript is “An optical technique to produce embedded quantum structures in semiconductors,” which we have successfully demonstrated. A device-level discussion lies outside the scope of this manuscript but will be the subject of our further publications.

To emphasize potential practical implications, we have added the following text on p. 12:

To conclude, the all-optical technique allows one to create sub-surface periodic potentials near a QW in a wide variety of semiconductors if regions containing deep-level impurities are introduced at certain distances from the QW. It is also worth mentioning that our experiments, as well as the experiments by Weiss et al [12-15], were conducted on non-planar Hall bar mesa-structure devices prepared by standard lithography. Importantly, because the height/depth of the non-planar surface structures was very small compared to the wavelength of the illumination light, they did not have any noticeable effect on the interference pattern inside the semiconductor samples. During the current and previous experiments, the technique was applied to devices that had fully opaque Ohmic contact areas. This did not affect the measurements thanks to using the standard four-probe technique. Moreover, surface electrodes can be made of transparent materials, such as standard indium-tin-oxide (ITO) conducting films.

On behalf of all Authors

Sergei Studenikin and Cyril Hnatovsky,                              April 30, 2023

Round 2

Reviewer 2 Report

Thanks the Authors' reply to clarify the novelty of this manuscript with their published J. Appl. Phys. 132, 044301 (2022).

For the technical part, frankly, I am not fully persuaded, especially on the deep level traps, on the metal electrode/ITO electrode.. but I accepted the author’s explanation.

Do hope to see more practical applications of this optical technique.